# Minimum Message Length Inference of the Exponential Distribution with Type I Censoring

**DOI:** 10.3390/e23111439

**Published:** 2021-10-30

**Authors:** Enes Makalic, Daniel Francis Schmidt

**Affiliations:** 1Melbourne School of Population and Global Health, The University of Melbourne, Parkville, VIC 3010, Australia; 2Faculty of Information Technology, Monash University, Clayton, VIC 3168, Australia; daniel.schmidt@monash.edu

**Keywords:** minimum message length, exponential distribution, maximum likelihood, survival analysis, censoring

## Abstract

Data with censoring is common in many areas of science and the associated statistical models are generally estimated with the method of maximum likelihood combined with a model selection criterion such as Akaike’s information criterion. This manuscript demonstrates how the information theoretic minimum message length principle can be used to estimate statistical models in the presence of type I random and fixed censoring data. The exponential distribution with fixed and random censoring is used as an example to demonstrate the process where we observe that the minimum message length estimate of mean survival time has some advantages over the standard maximum likelihood estimate.

## 1. Introduction

In Type I random censoring we observe for each item *i* either the true survival time $T_{i}=t_{i}$ ($t_{i}>0$) or the censoring time $C_{i}=c_{i}$ ($c_{i}>0$), where capital letters are used to denote random variables. The data consists of joint realisations of the random variables $\left(Y_{i}=y_{i},\Delta_{i}=\delta_{i}\right)$ (i=1,…,n) where
(1)$$\begin{array}{cc} Y_{i} & =\min(T_{i},C_{i}), \end{array}$$
(2)$$\begin{array}{cc} \Delta_{i} & =I\left(T_{i}\leq C_{i}\right)=\left\{\begin{array}{cc} 1, & \mathrm{if}\;T_{i}\leq C_{i}\;\left(\mathrm{observed}\;\mathrm{survival}\right)\\ 0, & \mathrm{if}\;T_{i}>C_{i}\;\left(\mathrm{observed}\;\mathrm{censoring}\right). \end{array}\right. \end{array}$$
The censoring time $C_{i}$ may be fixed (i.e., $C_{i}=c$ for all i=1,…,n) or a random variable that may depend on other factors (e.g., loss to follow-up). The likelihood function of *n* observed data points $D=\{\left(y_{1},\delta_{1}\right),\ldots ,\left(y_{n},\delta_{n}\right)\}$ is
$$p\left(D\right)=\prod_{i=1}^{n}{\left(p_{T}\left(y_{i}\right)\left(1-F_{C}\left(y_{i}\right)\right)\right)}^{\delta_{i}}{\left(p_{C}\left(y_{i}\right)\left(1-F_{T}\left(y_{i}\right)\right)\right)}^{1-\delta_{i}}$$
where $p_{T}\left(t|\theta \right)$ and $F_{T}\left(t|\theta \right)$ denote the probability density and the cumulative density function of the random variable *T*, respectively. Inference about the survival times $\left(t_{1},\ldots ,t_{n}\right)$ is of key interest in many areas of science and is commonly done by maximizing the likelihood and dropping terms relevant to *C* only.

This manuscript examines inference of models in the presence of censored data under the minimum message length (MML) framework. MML (see Section 3) is Bayesian technique for model selection and parameter estimation that is based on data compression and key principles of information theory. MML is known to possess strong theoretical properties [1,2,3] and has previously been successfully applied to a wide range of statistical models [1]. Here, we demonstrate how MML can be used to infer models under fixed censoring as well as type I random censoring. We use the exponential distribution (see Section 2) as a simple example to demonstrate the key steps and compare the MML estimator to the well-known maximum likelihood estimator in this setting (see Section 2.1). Although MML analysis of the exponential distribution is not new (see, for example, [1,4]), the MML principle has not been applied to any kind of survival data with censoring to date.

The main contributions of this manuscript are to: (i) introduce the MML principle of inductive inference and demonstrate how the Wallace–Freeman MML approximation can used to infer exponential models with type I censored data; (ii) show that the MML estimate of the mean lifetime has some advantages over the usual maximum likelihood estimate for small samples and that it converges to the maximum likelihood estimate for large sample sizes, (iii) incorporate the proposed codelengths for censored exponential distributions into MML finite mixture models allowing for inference of all parameters as well as the number of mixture classes; and (iv) compare the MML principle to the closely related minimum description length principle.

## 2. Exponential Distribution

Consider the case of a randomly censored exponential parameter studied in [5] where the lifetime data and the censoring data are assumed to be exponentially distributed
(3)$$T_{i}∼\mathrm{Exp}\left(\beta \right),\;C_{i}∼\mathrm{Exp}\left(\alpha \right),\;i=1,\ldots ,n,$$
and α,β>0 denote the mean censoring time and survival time, respectively. Under this model, the joint probability distribution of $\left(Y_{i}=y_{i},\Delta_{i}=\delta_{i}\right)$ is
(4)$$\begin{array}{ccc} p(Y_{i}=y_{i},\Delta_{i}=1) & = & p_{T}\left(y_{i}\right)\left(1-F_{C}\left(y_{i}\right)\right) \end{array}$$
(5)$$\begin{array}{ccc} p(Y_{i}=y_{i},\Delta_{i}=0) & = & p_{C}\left(y_{i}\right)\left(1-F_{T}\left(y_{i}\right)\right) \end{array}$$
where $\left(Y_{i},\Delta_{i}\right)$ are defined in (Equation 1) and (Equation 2) respectively. In contrast to random censoring, in fixed censoring an item is observed for a period of time, say c>0, and its actual survival time $t_{i}$ is known if the item fails before time $t_{i}\leq c$; otherwise, we only know that the item survived past $t_{i}>c$. In the case of exponentially distributed survival times, the observed data ($Y=y_{i},\Delta_{i}=\delta_{i})$ follows
(6)$$T_{i}∼\mathrm{Exp}\left(\theta \right),\;C_{i}=c,\;i=1,\ldots ,n,$$
where c>0 is a fixed constant (the follow up period) known a priori. Given *n* data points $D=\{\left(y_{1},\delta_{1}\right),\ldots ,\left(y_{n},\delta_{n}\right)\}$, the aim is to estimate the unknown mean lifetime survival β>0 (random censoring) or θ>0 (fixed censoring).

### 2.1. Maximum Likelihood Estimation

The method of maximum likelihood is the most common approach used to obtain parameter estimates in parametric models. Under the censored exponential model, maximum likelihood proceeds by setting the parameter estimate $\hat{\beta }\left(D\right)$ to the value that maximises the probability of the data. From (Equation 4) and (Equation 5), the joint probability of the data *D* is
(7)$$\begin{array}{cc} p(D|\alpha ,\beta )\; & ={\left(\frac{1}{\beta }\right)}^{k}{\left(\frac{1}{\alpha }\right)}^{n-k}\exp\left(-\left(\frac{1}{\beta }+\frac{1}{\alpha }\right)\sum_{i=1}^{n}y_{i}\right), \end{array}$$
where $k=(\sum_{i}\delta_{i})$ is the number of observed uncensored survival times. Maximizing the likelihood function is equivalent to minimizing the negative log-likelihood function
(8)$$-\log p\left(D|\alpha ,\beta \right)=k\log\beta +\left(n-k\right)\log\alpha +\left(\frac{1}{\beta }+\frac{1}{\alpha }\right)\sum_{i=1}^{n}y_{i}.$$
Maximum likelihood estimates of the mean survival and censoring times are
(9)$$\hat{\beta }{\left(D\right)}_{\mathrm{ML}}=\frac{1}{k}\sum_{i=1}^{n}y_{i},\;\hat{\alpha }{\left(D\right)}_{\mathrm{ML}}=\frac{1}{n-k}\sum_{i=1}^{n}y_{i},$$
respectively. Provided the count of observed survival times k∈(0,n), the maximum likelihood estimates $\hat{\alpha }\left(D\right)$ and $\hat{\beta }\left(D\right)$ are finite; otherwise, if k=0 or k=n, one of the maximum likelihood estimates $\hat{\alpha }\left(D\right)$ or $\hat{\beta }\left(D\right)$ is infinite. Kim [5] showed that maximum likelihood estimates have infinite mean and variance in this setting. However, the expected value of the maximum likelihood estimate $\hat{\beta }\left(D\right)$ is finite if we condition on k>0. Kim [5] further showed that, provided k∈(0,n), the maximum likelihood estimates $\hat{\alpha }\left(D\right)$ and $\hat{\beta }\left(D\right)$ are unbiased, strongly consistent (without any condition on *k*) and asymptotically normally distributed.

In the case of Type I censored data with a fixed censoring time c>0, the negative log-likelihood function of the data is
(10)$$-\log p\left(D|\theta ;c\right)=k\log\left(\theta \right)+\frac{1}{\theta }\left(\sum_{i=1}^{n}y_{i}\delta_{i}\right)+\frac{c(n-k)}{\theta }$$
where $k=\sum_{i}\delta_{i}$ as before. Under fixed censoring, the maximum likelihood estimate of the mean survival time $\hat{\theta }\left(D\right)$ is (see, for example, [6])
(11)$$\hat{\theta }\left(D\right)=\frac{c\left(n-k\right)+\sum_{i=1}^{n}\delta_{i}y_{i}}{k}.$$
In case of no censoring (i.e., k=n implying complete data), (Equation 11) reduces to $(\sum_{i}y_{i})/n$, which is the usual maximum likelihood estimate for the exponential distribution with complete data. The sampling distribution of (Equation 11) is asymptotically normal with mean θ and variance
(12)$$\frac{\theta^{2}}{n(1-\exp(-c/\theta ))}=\frac{\theta^{2}}{nF_{T}\left(c|\theta \right)}.$$
Conditional upon k>0, Mendenhall and Lehman [7] obtained the exact mean and variance of the maximum likelihood estimate (Equation 11)
$$E\left\{\hat{\theta }\right\}=\theta -c\left(\frac{q}{p}-nE\left\{k^{-1}\right\}+1\right),\;V\left\{\hat{\theta }\right\}={\left(nc\right)}^{2}V\left\{k^{-1}\right\}+\left(\theta^{2}-c^{2}q/p^{2}\right)E\left\{k^{-1}\right\},$$
where
p=1−exp(−c/θ),q=1−p,E{k−a}=11−qn∑k=1n1kankpkqn−k,
for a=1,2,… and $V\left\{k^{-1}\right\}=E\left\{k^{-2}\right\}-{\left(E\left\{k^{-1}\right\}\right)}^{2}$. However, the large sample normal approximation of the distribution of the maximum likelihood estimates is inaccurate and not representative of the behaviour of the estimate in the small to moderate sample size regime [7]. Balakrishnan and Davies [8] further show that the maximum likelihood estimate computed based on a censoring time $c^{\prime }$ will always produce an estimate which is Pitman closer to the data generating model θ than the maximum likelihood estimate computed with a shorter censoring time $c<c^{\prime }$. In the next section, we introduce the MML principle of inductive inference (see Section 3) and demonstrate how MML can be used to infer exponential models with censoring (see Section 4).

## 3. Minimum Message Length

Introduced in the late 1960s by Wallace and Boulton [9], the minimum message length (MML) principle [1,9,10,11] is a framework for inductive inference based on ideas in information theory and data compression. Under the MML framework, the aim is to transmit a set of data (a message) from a hypothetical sender to a receiver over a noiseless transmission channel. The MML message is designed to consist of two parts:the *assertion*: an encoding of the model structure and the associated model parameters $\theta \in \Theta \in R^{p}$.the *detail*: a description of the data *D* using the model p(D|θ) that was specified in the assertion.
The length of the assertion measures the complexity of the model, with complex models requiring longer codelengths compared to simpler models, while the detail captures how well a model fits the data. The length of the two-part message, I(D,θ), is the sum of the length of the assertion, I(θ), and the length of detail, I(D|θ); namely,
(13)$$I\left(D,\theta \right)=\underset{\mathrm{assertion}}{\underset{︸}{I(\theta )}}+\underset{\mathrm{detail}}{\underset{︸}{I(D|\theta )}}.$$
Within the MML framework we seek the model
(14)$$\hat{\theta }\left(D\right)=\underset{\theta \in \Theta }{arg min}\left\{I(D,\theta )\right\}$$
that minimises the length of this message. Due to the two-part nature of the message, MML automatically balances the trade-off between model complexity and the goodness of fit of the model to the data. By measuring the quality of a model in (say) bits, MML is a yardstick that can be universally used to compare models with different parameters and structures.

There exist several approches to computing message lengths (Equation 13), with the strict MML procedure (SMML) [1,12] and the MML87 approximation [1,10] being the most widely known. In contrast to the SMML procedure whose construction is known to be NP hard [13], the MML87 approximation is computationally tractable and most widely used in practice. The MML87 codelength approximation to (Equation 13) is
(15)$$I_{87}\left(D,\theta \right)=\underset{\mathrm{assertion}}{\underset{︸}{-\log\pi \left(\theta \right)+\frac{1}{2}\log\left|J_{\theta }\left(\theta \right)\right|+\frac{p}{2}\log\kappa_{p}}}+\underset{\mathrm{detail}}{\underset{︸}{\frac{p}{2}-\log p\left(D|\theta \right)}}$$
where $\pi_{\theta }\left(\theta \right)$ is the prior distribution for the parameters θ, $|J_{\theta }\left(\theta \right)|$ is the determinant of the expected Fisher information matrix, p(D|θ) is the likelihood function of the model and $\kappa_{p}$ is a quantization constant [14,15] that depends on the number of parameters *p*. Specifically, for small *p* we have
(16)$$\kappa_{1}=\frac{1}{12},\;\kappa_{2}=\frac{5}{36\sqrt{3}},\;\kappa_{3}=\frac{19}{192\times 2^{1/3}},$$
while $\kappa_{p}$ is well-approximated for large *p* by [1]:(17)$$\frac{p}{2}\left(\log\kappa_{p}+1\right)\approx -\frac{p}{2}\log2\pi +\frac{1}{2}\log p\pi -\gamma ,$$
where γ≈0.5772 is the Euler–Mascheroni constant. The MML87 codelength, evaluated at the minimum, is the shortest codelength of a two-part message that encodes both the model parameters θ∈Θ and the data *D*. The MML87 approximation is known to be invariant under smooth one-to-one reparameterizations of the likelihood function and is asymptotically equivalent to the well-known Bayesian information criterion (BIC) [16] as n→∞ with p>0 fixed; that is,
(18)$$I_{87}\left(D,\theta \right)=-\log p\left(D|\theta \right)+\frac{p}{2}\log n+O\left(1\right)$$
where the O(1) term depends on the prior distribution, the Fisher information and the number of parameters *p*. Unlike model selection criteria such as Akaike’s information criterion (AIC) and BIC, MML allows for both parameter estimation and model selection within the same unified framework. Furthermore, in models where the number of parameters grows with *n* or the sample size is relatively small, the difference between the MML87 codelength and BIC can be substantial. Examples include analysis of multiple short time series, where several measurements are collected over a period of time for a large number of study participants [17], learning finite mixture models [18] and discriminating between Poisson and geometric distributions based on observed data [19]. In the latter example, both the Poisson and geometric distribution have the same number of free parameters so that model selection with BIC is equivalent to choosing the model with the higher likelihood. In contrast, MML87 takes into account the complexity of each distribution [20] and not just the number of parameters, resulting in improved model selection performance for small sample sizes [19].

MML has been successful applied to a wide range of problems (e.g., decision trees [21], factor analysis [22], linear causal models [23], mixture modelling [18,24]) demonstrating excellent parameter estimation properties and model selection performance that is on par or better than commonly used techniques such as Akaike’s information criterion (AIC) [25] and the Bayesian information criterion (BIC). A brief tutorial overview of minimum message length can be found in [19].

## 4. Minimum Message Length Inference of Type I Censored Exponential Data

To encode and transmit censored data $D=\{\left(y_{1},\delta_{1}\right),\ldots ,\left(y_{n},\delta_{n}\right)\}$ between the hypothetical sender and receiver within the MML framework, we have two options:Transmit the censoring indicators $\left(\delta_{1},\ldots ,\delta_{n}\right)$ first and then transmit the lifetime survival data $\left(y_{1},\ldots ,y_{n}\right)$ given the receiver now knows which of the *n* data points are censored (see Section 4.1);Transmit the censoring indicators and the lifetime data simultaneously (see Section 4.2).
We shall now estimate the MML87 codelength (Equation 15) for both the joint and the conditional encoding schemes for the censored exponential distribution setting introduced in Section 2.

### 4.1. Conditional Encoding of the Data

Under the conditional encoding framework, the sender transmits the censoring indicators $\delta_{i}$ first, and then transmits the lifetime data $y_{i}$ using the conditional distribution of the data given the observed censoring indicators. The total message length of the data $D=\{\left(y_{1},\delta_{1}\right),\ldots ,\left(y_{n},\delta_{n}\right)\}$ and the parameters θ with the conditional encoding is
(19)$$I_{87}\left(D,\theta \right)=I_{87}\left(\phi ,\delta \right)+I_{87}\left(\psi ,y|\delta \right),$$
where θ={ϕ,ψ} are the model parameters defined below, I(ϕ,δ) denotes the message length of the censoring indicators $\delta =(\delta_{1},\ldots ,\delta_{n})$, and I(ψ,y|δ) denotes the codelength of the survival data $y=(y_{1},\ldots ,y_{n})$, given that the censoring indicators are known to the receiver. From (Equation 3), the probability of observing an uncensored datum, say ϕ>0, is
(20)$$\phi =P\left(T_{i}\leq C_{i}\right)=\frac{\alpha }{\alpha +\beta },\;\left(i=1,\ldots ,n\right),$$
implying that the censoring indicators follow a Bernoulli distribution with probability ϕ; that is, $\delta_{i}∼\mathrm{Bernoulli}\left(\phi \right)$, or equivalently, *k* follows the binomial distribution k∼binomial(ϕ,n).

The MML87 codelength of the binomial distribution was previously derived in [1,18] and is included here for completeness. Briefly, to compute the MML87 codelength (Equation 15) we require the Fisher information $J_{\phi }\left(\phi \right)$ and the prior distribution $\pi_{\phi }\left(\phi \right)$ for the probability of observing an uncensored datum. The Fisher information is well-known
(21)$$J_{\phi }\left(\phi \right)=\frac{n}{\phi (1-\phi )}.$$
We assume the prior distribution for the censoring probability ϕ to be the beta distribution (ϕ∼beta(a,b)) with probability density function
(22)$$\pi_{\phi }\left(\phi |a,b\right)=\frac{\phi^{a-1}{\left(1-\phi \right)}^{b-1}}{B(a,b)}$$
where a,b>0 are the shape and scale parameters respectively and B(a,b) is the usual beta function. Substituting (Equation 21) and (Equation 22) into the MML87 codelength (Equation 15) and noting that $\kappa_{1}=1/12$, yields
(23)$$I_{87}\left(\phi ,\delta \right)=-\left(k+a-\frac{1}{2}\right)\log\phi -\left(n+b-\frac{1}{2}-k\right)\log\left(1-\phi \right)\;+\log B\left(a,b\right)+\frac{1}{2}\left(1+\log n-\log12\right)$$
where, as before, $k=(\sum_{i}\delta_{i})$. The codelength (Equation 23) is minimised at the MML87 estimate
(24)$${\hat{\phi }}_{87}\left(\delta \right)=\frac{k+a-1/2}{n+a+b-1}.$$
Note that, in the special case of uniform prior distribution (a=b=1), the MML87 estimate simplifies to
(25)$${\hat{\phi }}_{87}\left(\delta \right)=\frac{k+1/2}{n+1}.$$
The shortest MML87 codelength for the censoring indicators is therefore given by $I_{87}\left({\hat{\phi }}_{87},\delta \right)$. It remains to work out the conditional codelength of the surivival times given the censoring indicators, $I_{87}\left(\psi ,y|\delta \right)$.

We note that the conditional likelihood of the lifetime datum $y_{i}$ is
(26)$$p\left(y_{i}|\alpha ,\beta ,\delta =0\right)=p\left(y_{i}|\alpha ,\beta ,\delta =1\right)=\left(\frac{1}{\beta }+\frac{1}{\alpha }\right)\exp\left(-y_{i}\left(\frac{1}{\beta }+\frac{1}{\alpha }\right)\right),$$
which is the exponential distribution with mean $\psi ={\left(1/\beta +1/\alpha \right)}^{-1}$; that is,
(27)$$y_{i}|\delta_{i}∼\mathrm{Exp}\left(\psi \right),\;i=1,\ldots ,n.$$
The Fisher information of the exponential distribution is
(28)$$J_{\psi }\left(\psi \right)=\frac{n}{\psi^{2}}.$$
In terms of the prior distribution for ψ, Schmidt and Makalic [4] consider the conjugate exponential distribution with a hyperparameter $\psi_{0}$ that controls the prior mean. Here, we would like an objective prior distribution on the mean ψ that is free of hyperparameters and has heavy tails so that large values of ψ are not penalized too severely. Additionally, our choice of the prior distribution should ideally lead to an easy to compute analytic estimate of ψ. A reasonable option is the half-Cauchy distribution which has heavy tails however it leads to MML estimates that are roots of polynomial functions of $s=\sum_{i}\left(y_{i}\right)$. Instead, we will use the Fréchet (inverse Weibull) distribution with probability density function
(29)$$\pi_{\psi }\left(\psi \right)=\psi^{-2}\exp\left(-\psi^{-1}\right),\;\psi >0,$$
which is a type of generalized extreme value distribution and has Cauchy-like heavy tails. Substituting (Equation 29), (Equation 28) into (Equation 3), we obtain the MML87 codelength
(30)$$I_{87}\left(\psi ,y|\delta \right)=\left(n+1\right)\log\psi +\frac{1}{\psi }\left(1+\sum_{i=1}^{n}y_{i}\right)+\frac{1}{2}\left(1+\log n-\log12\right).$$
The MML87 estimate of the mean ψ is
(31)$${\hat{\psi }}_{87}\left(y\right)=\frac{s+1}{n+1}$$
where, as before, $s=\sum_{i}y_{i}$. The MML87 estimate corresponds to the usual maximum likelihood estimate ${\hat{\psi }}_{\mathrm{ML}}\left(y\right)=s/n$ with one additional data point that has a unit contribution to the mean. The expected mean squared error of the MML87 estimate is
(32)$$E\left\{{\left({\hat{\psi }}_{87}\left(y\right)-\psi \right)}^{2}\right\}=\frac{\psi (\psi (n+1)-2)+1}{{\left(n+1\right)}^{2}}$$
which dominates the maximum likelihood estimate for
(33)$$\psi >\frac{n}{n+\sqrt{n(2n+1)}},\;n>0.$$
As the sample size *n* increases, we note that
(34)$$\lim_{n\to \infty }\left\{\frac{n}{n+\sqrt{n(2n+1)}}\right\}=\sqrt{2}-1\approx 0.414$$
implying the MML87 estimate dominates maximum likelihood for all ψ>0.414 in terms of expected mean squared error for large *n*. However, we note that, unlike the MML87 estimate with this choice of prior distribution, the maximum likelihood estimate is invariant to scaling of the data.

Substituting $I_{87}\left(\hat{\phi },\delta \right)$ and $I_{87}\left(\hat{\psi },y|\delta \right)$ into (Equation 19) yields the total (conditional) codelength of the data. The MML87 estimates of the mean lifetime ${\hat{\beta }}_{87}\left(D\right)$ and censoring time ${\hat{\alpha }}_{87}\left(D\right)$ can be recovered from
(35)$$\alpha \to \frac{\psi }{1-\phi },\;\beta \to \frac{\psi }{\phi },$$
for ϕ∈(0,1). Next, we examine how the same message can be encoded using joint encoding of lifetime data and censoring indicators.

### 4.2. Joint Encoding of the Data

Unlike in the conditional encoding, the sender now transmits the survival times and the indicator variables simultaneously. The negative log-likelihood function of the data $D=\{\left(y_{1},\delta_{1}\right),\ldots ,\left(y_{n},\delta_{n}\right)\}$ is given in (Equation 8). The Fisher information in this parameterization is
(36)$$J\left(\alpha ,\beta \right)=\frac{n^{2}}{\beta \alpha {\left(\alpha +\beta \right)}^{2}}.$$
We would like to use prior distributions for α and β that are comparable to those in the conditional coding described in Section 4.1. Noting that ϕ∼beta(a,b), ψ has the standard Frechet distribution and
(37)$$\phi =\frac{\alpha }{\alpha +\beta },\;\psi ={\left(\frac{1}{\alpha }+\frac{1}{\beta }\right)}^{-1},$$
the Jacobian of the transformation from (ϕ,ψ)→(α,β) is
(38)$$\frac{\alpha \beta }{{\left(\alpha +\beta \right)}^{3}}$$
implying that a commensurate joint prior distribution for α,β is
(39)$$\pi_{\alpha ,\beta }\left(\alpha ,\beta \right)=\frac{\alpha^{a-2}e^{-\frac{\alpha +\beta }{\alpha \beta }}\beta^{b-2}{\left(\alpha +\beta \right)}^{-a-b+1}}{B(a,b)}$$
where B(·,·) is the beta function. In the special case where ϕ is given a uniform prior (i.e., a=b=1), we have
(40)$$\pi_{\alpha ,\beta }\left(\alpha ,\beta \right)=\frac{e^{-\frac{\alpha +\beta }{\alpha \beta }}}{\alpha \beta (\alpha +\beta )}$$
Substituting (Equation 36) and (Equation 39) into (Equation 15), the MML87 codelength is
(41)$$\begin{array}{cc} I_{87}\left(D,\theta \right) & =k\log\beta +\left(n-k\right)\log\alpha +\left(\frac{1}{\alpha }+\frac{1}{\beta }\right)\left(\sum_{i=1}^{n}y_{i}\right)-\log\pi_{\alpha ,\beta }\left(\alpha ,\beta \right)\\ \; & +\log n-\frac{1}{2}\log\left(\alpha \beta {\left(\alpha +\beta \right)}^{2}\right)+\log\kappa_{2}+1 \end{array}$$
where θ={α,β} and the quantization constant $\kappa_{2}=5/\left(36\sqrt{3}\right)$. The MML87 estimates that minimize the codelength (Equation 41) are
(42)$$\begin{array}{cc} {\hat{\alpha }}_{87}\left(D\right) & =\frac{2(s+1)(a+b+n-1)}{\left(n+1)(2b-2k+2n-1\right)},\;{\hat{\beta }}_{87}\left(D\right)=\frac{2(s+1)(a+b+n-1)}{\left(n+1)(2a+2k-1\right)}. \end{array}$$
If required, the corresponding estimates of ϕ and ψ can be obtained from (Equation 37).

### 4.3. Properties

First we show the the conditional (Equation 19) and joint (Equation 41) MML codelengths are equivalent up to a constant to be specified below. From Section 4.1 and Section 4.2, we note the joint density of (Y,Δ) can be expressed as a product of the binomial $p_{\Delta }\left(\delta |\phi \right)$ and exponential densities $p_{Y}\left(y|\psi \right)$
$$p_{Y,\Delta }\left(y,\delta |\alpha ,\beta \right)=p_{\Delta }\left(\delta |\phi \right)p_{Y}\left(y|\psi \right)=\left(\phi^{\delta }{\left(1-\phi \right)}^{1-\delta }\right)\left(\exp\left(-y/\psi \right)/\psi \right)$$
where *Y* and Δ are independent random variables (see Kim [5] (p. 104)). Consequently, as MML87 is invariant under smooth one-to-one reparameterizations of the sampling model, the MML87 joint codelength (Equation 41) and the corresponding conditional codelength (Equation 19) are identical (except for the minor efficiency gain in the joint codelength discussed below). Specifically, the relationship between the joint codelength, $I_{87}\left(\alpha ,\beta ,D\right)$ and conditional codelength, $I_{87}\left(\psi ,\phi ,D\right)$, can be expressed as
(43)$$I_{87}\left(\psi ,\phi ,D\right)=I_{87}\left(\alpha ,\beta ,D\right)+\log\left(\frac{3\sqrt{3}}{5}\right)$$
where the term $\log(3\sqrt{3}/5)\approx 0.0385$ arises due to the quantization constant being smaller in higher dimensions since its more efficient to encode multiple parameters simultaneously compared to encoding each parameter independently.

Furthermore, as the MML87 estimate of ϕ is $\hat{\phi }\in \left(0,1\right)$ (see (Equation 24)), MML87 estimates of the mean survival times (α,β) are finite for all k∈[0,n] in contrast to the corresponding maximum likelihood estimates (Equation 9) which are finite for k∈(0,n). As n→∞, it is well-known that the MML87 estimates are equivalent to the maximum likelihood estimates (see (Equation 18)) which implies that the MML87 estimates are similarly asymptotically normally distributed and strongly consistent.

The expected mean square error $E\{{\left(\hat{\beta }-\beta^{*}\right)}^{2}\}$ of the ML and MML87 estimates of $\beta^{*}$, conditional on k>0, is expressible in terms of the generalized hypergeometric function for any n>0. Figure 1 (top) depicts the expected mean squared error between the MML87 and ML estimate of $\beta^{*}$, expressed as a ratio of MML87 to ML with smaller values indicating preference for the MML87 estimate. The expected mean squared error of the MML87 estimate of $\beta^{*}$ was generally lower than the corresponding maximum likelihood estimate (except when the true censoring proportion $\phi^{*}$ was small) with the biggest difference observed for small sample sizes, while the two estimates were practically indistinguishable for larger sample sizes n≥100.

We also compared the MML87 and the ML estimates in terms of the relative entropy or the Kullback–Leibler (KL) divergence, conditional on k∈(0,n). The KL divergence between the true data generating model $\left(\alpha_{1},\beta_{1}\right)$ and the approximating model $\left(\alpha_{2},\beta_{2}\right)$ is
$$D_{\mathrm{KL}}(\alpha_{1},\beta_{1}\left|\right|\alpha_{2},\beta_{2})=\frac{\alpha_{1}\beta_{1}\left(\alpha_{2}+\beta_{2}\right)+\alpha_{2}\beta_{2}\left(\alpha_{1}\log\left(\frac{\beta_{2}}{\beta_{1}}\right)+\beta_{1}\log\left(\frac{\alpha_{2}}{\alpha_{1}}\right)-\alpha_{1}-\beta_{1}\right)}{\alpha_{2}\beta_{2}\left(\alpha_{1}+\beta_{1}\right)},$$
which, as expected, is the sum of the KL divergences between two exponential and two binomial distributions. The KL divergence may be interpreted as the expected amount of extra information required to encode data from $\left(\alpha_{1},\beta_{1}\right)$ using the model $\left(\alpha_{1},\beta_{2}\right)$. The expected KL divergence for both the ML and MML estimators is shown in the bottom of Figure 1, conditional on k>0 and k<n. It is clear that for n=5 the MML87 estimate dominates the maximum likelihood estimate in terms of the KL divergence for all $\phi^{*}\in \left(0.05,0.95\right)$. When the sample size is increased (n=10), the MML87 estimate exhibits smaller KL divergence compared to the ML estimate for all $\phi^{*}$ except when $\phi^{*}\to 0$ or $\phi^{*}\to 1$ where the maximum likelihood estimate has smaller KL divergence.

## 5. Minimum Message Length Inference with Fixed Censoring

Consider now the fixed censoring scenario (Equation 6) introduced in Section 2 where the negative log-likelihood function of the data is given in (Equation 10). If we wish to encode the data *D* using the joint MML code (see Section 4.2), we require the negative log-likelihood, the Fisher information and a prior distribution for the mean survival time θ>0. The negative log-likelihood is given in (Equation 10) while the Fisher information for Type I censored data with fixed censoring is:(44)$$J_{\theta }\left(\theta ;c\right)=\frac{n(1-\exp(-c/\theta ))}{\theta^{2}}=\frac{nF_{T}\left(c|\theta \right)}{\theta^{2}},$$
where $F_{T}\left(\cdot |\theta \right)$ is the cumulative distribution function of the survival data *T* (see Section 2). The reduction in information due to censoring is clearly a function of θ and the cumulative density function of *T*, with large *c* resulting in little information loss compared to small *c*. As expected, as *c* gets larger
(45)$$\lim_{c\to \infty }J_{\theta }\left(\theta ;c\right)=\frac{n}{\theta^{2}},$$
which is the usual Fisher information for the exponential distribution with no censoring. The prior distribution for θ is chosen to be the Fréchet ditribution with scale *c* and probability density function
(46)$$\pi_{\theta }\left(\theta ;c\right)=c^{-1}{\left(\frac{\theta }{c}\right)}^{-2}\exp\left(-\frac{c}{\theta }\right).$$
Substituting (Equation 44) and (Equation 46) into (Equation 15), we obtain the complete MML87 codelength for the joint encoding
(47)$$\begin{array}{cc} I_{87}\left(D,\theta \right)\; & =\left(k+1\right)\log\left(\theta \right)+\frac{1}{\theta }\left(c\left(\left(n-k\right)+1\right)+\sum_{i=1}^{n}y_{i}\delta_{i}\right)\\ \; & +\frac{1}{2}\log\left(\frac{1-\exp(-c/\theta )}{c^{2}}\right)+\frac{1}{2}\left(1+\log n-\log12\right). \end{array}$$
Due to the form of the Fisher information, the MML87 estimate of θ that minimizes this codelength is unavailable analytically and must be obtained via numerical optimisation. The maximum likelihood estimate (Equation 11) may be used as a starting point for the numerical search.

Consider now the conditional encoding (see Section 4.1) where the probability of observing an uncensored datum, say ϕ>0, is
(48)$$\phi =P\left(T_{i}\leq c\right)=F_{T}\left(c|\theta \right)=1-\exp\left(-c/\theta \right),\;\left(i=1,\ldots ,n\right),$$
so that the number of uncensored data points *k* follows the binomial distribution k∼binomial(ϕ,n). This implies that the mean survival time can then written as
(49)θ=−c/log(1−ϕ),ϕ∈(0,1).
A naive conditional coding approach proceeds by encoding the censoring indicators following Section 4.1 with codelength (Equation 23). To encode I(y|δ), one would use the conditional probabilities of the lifetime data which are
(50)$$\begin{array}{cc} p_{Y|\Delta }\left(Y_{i}=y_{i}|\Delta_{i}=1\right)\; & =\frac{p(T_{i}=y_{i})}{p(T_{i}\leq c)}\;\mathrm{if}\;y_{i}\leq c, \end{array}$$
and
(51)$$p_{Y|\Delta }\left(Y_{i}=c|\Delta_{i}=0\right)=1,\;\;p_{Y|\Delta }\left(Y_{i}>c|\Delta_{i}=0\right)=0.$$
for all i=1…,n. The conditional likelihood of the $k=(\sum_{i}\delta_{i})$ data points is then
(52)$$\begin{array}{cc} p(y|\theta ;\delta =1)\; & =\prod_{i:\delta_{i}=1}\frac{\left(1/\theta \right)\exp\left(-y_{i}/\theta \right)}{1-\exp(-c/\theta )}=-\prod_{i:\delta_{i}=1}\frac{{\left(1-\phi \right)}^{y_{i}/c}\log\left(1-\phi \right)}{c\phi }. \end{array}$$
Once the receiver has the censoring data and an estimate of ϕ, they implicitly know θ from (Equation 49). The length of the message required to transmit the data y is
(53)$$\begin{array}{cc} I_{87}\left(\theta ,y|\delta \right) & =k\log\theta +\frac{1}{\theta }\sum_{i:\delta_{i}=1}y_{i}+k\log\left(1-\exp\left(-c/\theta \right)\right)\\ \; & =-\frac{1}{c}\left(\sum_{i:\delta_{i}=1}y_{i}\right)\log\left(1-\phi \right)+k\log c\phi -k\log\left(-\log\left(1-\phi \right)\right) \end{array}$$
which is the negative log-likelihood of the data. However, this codelength is inefficient since the probability of censoring ϕ(θ) is not independent of the mean survival time θ. This implies that the precision to which ϕ(θ) is encoded must depend on the lifetime data y, which is not the case in the naive approach where the precision quantum for ϕ(θ) depends on the censoring data δ only. Consequently, joint MML coding should be used instead of the conditional encoding approach for the fixed censoring setup.

### 5.1. Example

We observe n=20 items with an exponential life distribution for c=150 h. Out of the 20 items k=15 items fail during the observation period and the sum of their lifetimes (in hours) is $s=\sum_{i}y_{i}\delta_{i}=835$ [6]. The maximum likelihood estimate of the mean lifetime θ is
(54)$${\hat{\theta }}_{\mathrm{ML}}\left(D\right)=\frac{150(5)+835}{15}=105.6\;h,$$
with a negative log-likelihood of 84.9 at the minimum. The MML87 estimate is obtained by a numerical search and is ${\hat{\theta }}_{87}\left(D\right)=110.1$ h with a codelength of 124.905 bits. The MML87 codelength at the maximum likelihood estimate is 124.924 bits suggesting that there is little difference between the two estimates in this example.

### 5.2. Properties

To evaluate the performance of the MML87 estimate, we computed the mean squared error risk and the expected Kullback–Leibler (KL) divergence for MML87 and the ML estimates under the data generating model $\theta^{*}=1$ and sample sizes n∈{5,25}. Since the ML estimate is undefined for k=0, all the results discussed below are conditional on k>0. The KL divergence from the ‘true’ model $\theta_{1}$ to the approximating model $\theta_{2}$ is
(55)$$D_{\mathrm{KL}}(\theta_{1}\left|\right|\theta_{2})=\left(1-\frac{\theta_{1}}{\theta_{2}}+\log\left(\frac{\theta_{1}}{\theta_{2}}\right)\right)\left(\exp\left(-\frac{c}{\theta_{1}}\right)-1\right).$$
The results are shown in Figure 2 where the *x*-axis of each plot is the censoring point *c* set to the *p*-th percentile of the data generating model Exp($\theta^{*}=1$); for example c=0.69 corresponds to the p=0.50-th percentile of Exp($\theta^{*}=1$). It is clear that the MML87 estimate is a reasonable alternative to the maximum likelihood estimate under fixed censoring. For p≥0.20 (i.e., the 20-th percentile of Exp($\theta^{*}=1$)) the MML87 estimate dominates the ML estimate in terms of KL risk, while for p<0.20 the estimates are very similar for both sample sizes tested. In terms of the expected mean squared error, the MML87 estimate dominates the ML estimate for all 0.20≤p≤0.80 when n=5; for n=25, the estimates are indistinguishable for p<0.1 and p>0.69 and the MML87 estimate again dominates ML for all 0.1<p<0.5.

## 6. Discussion

This manuscript has demonstrated how minimum message length can be used to infer data with censoring information. Specifically, we have derived MML87 codelengths for the exponential distribution with fixed censoring and random type I censoring. Although information theoretic universal models for the exponential distribution, including those corresponding to MML codes, are known [4], this is the first time MML has been applied to censored data.

The MML87 codelength for the exponential distribution with censoring provides a new means of parameter estimation as well as model selection. In terms of parameter estimation, the MML87 estimate of the mean lifetime θ under type I censoring described in this paper has some advantages over the usual maximum likelihood estimate for small sample sizes. First, the MML87 estimate is defined for all proportions of censoring unlike the maximum likelihood estimate which does not exist when all observations are censored; i.e., $k=(\sum_{i}\delta_{i})=0$. In addition, the MML87 estimate has on average lower mean squared error risk and lower KL divergence from the data generating model for a wide range of censoring proportions.

In the case of random censoring, the MML87 estimate is available in closed-form while for fixed censoring, it can only be obtained by numerical optimisation. Although the experiments in the manuscript utilised heavy-tailed prior distributions for the scale parameter as recommended in, for example, [26], the Bayesian nature of MML allows for information prior information to be incorporated directly into the estimation process. The effect of the prior distribution in the examples considered here is expected to be negligible for medium to large sample sizes.

Importantly, the proposed MML87 codelengths can also be used to discriminate between competing models (e.g., exponential vs lognormal) and offer some advantages over the well-known BIC model selection approach. BIC only considers the sample size and the number of parameters when measuring model complexity. In contrast, MML takes into account not just the number of parameters, but also the complexity of the distribution (i.e., the number of random data strings that are fitted well by the distribution). As the sample size n→∞, the MML87 codelength converges to the BIC and therefore inherits the favourable asymptotic properties of BIC, such as model selection consistency.

The codelengths derived in this manuscript are extendable to MML inference of other censored data types, such as the Weibull and the lognormal distribution, and can be incorporated into more complex models as shown in the next section.

### 6.1. Clustering Survival Data

To demonstrate the applicability of the codes derived in the manuscript, we implemented the MML87 codes into a Matlab software package for inference of finite mixture models. Our software, called Matlab Snob, features mixture models with categorical data (e.g., multinomial distribution), count data (e.g., geometric, Poisson and negative binomial distributions), continuous data (e.g., normal, Laplace, gamma and Weibull distributions) and survival data (type I fixed and random censored exponential distribution). As a demonstration of Matlab Snob, we used two publicly available survival data sets: (1) Rossi et al.’s criminal recidivism data [27], and (2) survival from malignant melanoma [28].

The crime data was recently analyzed in [29] using variational Bayes estimated finite mixture models. For clustering we used all n=432 observations and the following seven attributes: (1) financial aid (no, yes), (2) full-time work experience before incarceration (no, yes), (3) marital status at time of release (married, not married), (4) released on parole (no, yes), (5) number of convictions prior to current incarceration, (6) age in years at time of release and (7) week of first arrest after release (73.6% censored). This is an example of fixed censoring as all censored observations were censored at 52 weeks. We modelled the categorical attributes using a multinomial distribution, number of convictions was modelled with a negative binomial distribution, while a Gaussian distribution was used for age at time of release. For the week of arrest, we used the exponential distribution model with fixed type I censored data (see Section 5).

The melanoma data set consists of n=205 patients from Denmark who were diagnosed with malignant melanoma. Five attributes were used for clustering: (1) sex (male, female), (2) ulcer (present, absent), (3) age at diagnosis in years, (4) tumour thickness in mm, and (5) censored survival time in years (65.3% censored). Sex and ulcer were modelled via multinomial distributions, while age and tumour thickness were modelled with univariate Gaussian distributions. For the survival time, we used an exponential distribution with random type I censoring (see Section 4) and combined death due to melanoma and death due to other causes as the primary outcome of interest.

Clustering results for the Crime and the Melanoma data sets with Matlab Snob are shown in Table 1. First, since Matlab Snob learns finite mixture models using the MML87 codelength approximation, the same framework is used to estimate all model parameters as well as select the number of classes. For the Crime data, the model with three classes had the smallest codelength, while two classes were selected for the Melanoma data set. We observe that all the classes are relatively well differentiated in terms of average survival time. In the case of the Crime data set, class 1 had the shortest average time to arrest (θ=119 weeks) and consists of younger individuals (mean age 20.7 years, std. dev. 2.1 years) who are primarily unmarried and have no full-time work experience before incarceration. In contrast, class 3 comprised older individuals (mean age 36.3 years, std. dev. 4.9 years) 82% of which had full-time work experience, was estimated to have longest average time to arrest (θ=345.9 weeks). For the melanoma data set, class 1 was estimated to have the shortest average survival time (β=7.3 years) and consisted of individuals diagnosed at an older age (mean: 57.3 years, std. dev. 17.1 years) with larger tumours (mean: 5.4 mm, std. dev. 3.4 mm). In contrast, patients assigned to class 2 were diagnosed at a younger age, had smaller tumours and were estimated to have longer survival time on average (β=37.0 years).

The Matlab Snob clustering software is freely available for download from the Mathworks Filexchange website (ID: 72310) and will be extended to incorporate other survival distributions in the future (eg, Weibull and lognormal distribution). We note that the MML87 codelengths for type I censored exponentially distributed data derived in this paper can also be used in decision tree modelling [21,30,31]. For example, one could represent the leaves of the decision tree with a censored exponential distribution and use MML to infer an optimal decision tree for a data set.

### 6.2. Minimum Message Length and Minimum Description Length

Minimum message length is closely related to minimum description length (MDL), an inductive inference principle independently developed by Rissanen and colleagues [32,33,34,35]. Like MML, the MDL principle is rooted in information theory and, given a data set, seeks a model that would result in the shortest encoding of the data. A recent and popular version of the MDL principle is the normalized maximum likelihood (NML) code which says that the codelength for data y with respect to model class M parameterised by models $\theta \in R^{p}\in M$ is
(56)$$-\log p_{\mathrm{NML}}\left(y|M\right)=-\log p\left(y|\hat{\theta }\left(y\right),M\right)+\log\sum_{x}p\left(x|\hat{\theta }\left(x\right),M\right)$$
where $\hat{\theta }$ is the maximum likelihood estimate of the *p* parameters and the sum in the second term is taken over the entire data space; we replace the sum with an integral in the case of continuous data. The first term in the NML codelength is the negative log-likelihood of the data evaluated at the maximum likelihood estimate, while the second term represents the parametric complexity of the model class and measures how well models θ∈M within the model class M approximate random data sequences. In particular, a high parametric complexity says that a large number of data sequences can be well-approximated by models within the class. In contrast, the parametric complexity of a simple model that can only well-approximate a few data sequences will tend to be small.

Rissanen [33] derives an asymptotic approximation for the NML codelength which is accurate for medium to large sample sizes:(57)$$-\log p_{\mathrm{NML}}\left(y|M\right)=-\log p\left(y|\hat{\theta }\left(y\right),M\right)+\log\int_{\Theta }\sqrt{|J_{1}\left(\theta \right)|}+\frac{p}{2}\log\left(\frac{n}{2\pi }\right)+o\left(1\right)$$
where $J_{1}\left(\cdot \right)$ is the per-sample Fisher information matrix. Mera et al. [36] derive a somewhat sharper approximation to the NML codelength using Riemannian geometry tools and apply their new approximation to principal component analysis. Rissanen further shows that, like the MML87 codelength, the NML codelength reduces to the well-known Bayesian information criterion (BIC) in the limit as the sample size n→∞. Unfortunately, in the case of the exponential distribution with or without type I censoring, the parametric complexity is infinite for both the exact NML codelength and the asymptotic approximation. To circumvent the problem of infinite parametric complexity, one may consider the restricted approximate normalized maximum likelihood (ANML), the two-part ANML or the objective Bayesian code, among others [37].

Although there exist many similarities in the approaches to inference between MML and MDL, there are some important differences which we summmarize below:MDL relies on the maximum likelihood estimator and does not offer new means for parameter estimation;MDL is decidedly non-Bayesian avoiding the use of any (subjective or objective) prior information;MDL nominates the *model class* M that would result in the shortest encoding of the data and does not does not infer a fully specified model;while MML minimises the expected (average) codelength of the data with respect to the marginal data distribution, MDL minimizes the worst-case codelength relative to the ideal code.

In addition to the NML code, other MDL codes exist including the sequential NML code [38] and the conditional NML distribution [34], among others. Clearly, both MML and MDL approaches to inductive inference have merit, and if used correctly, will result in excellent model selection performance as shown in a wide range of applications. A more detailed discussion of MML and MDL similarities and differences can be found in [39] and [1] (pp. 413–415).

## Figures and Tables

**Figure 1 entropy-23-01439-f001:**
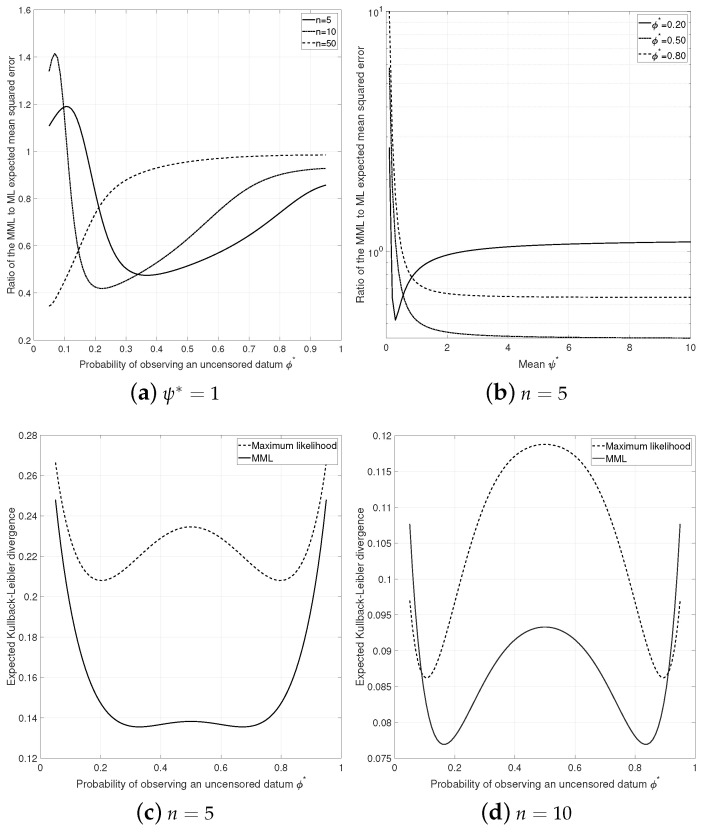
Expected mean squared error of $\beta^{*}$ and expected KL divergence between the MML87 and ML estimates. Ratio values less than 1 imply that the MML87 estimate has smaller mean squared error in estimating β.

**Figure 2 entropy-23-01439-f002:**
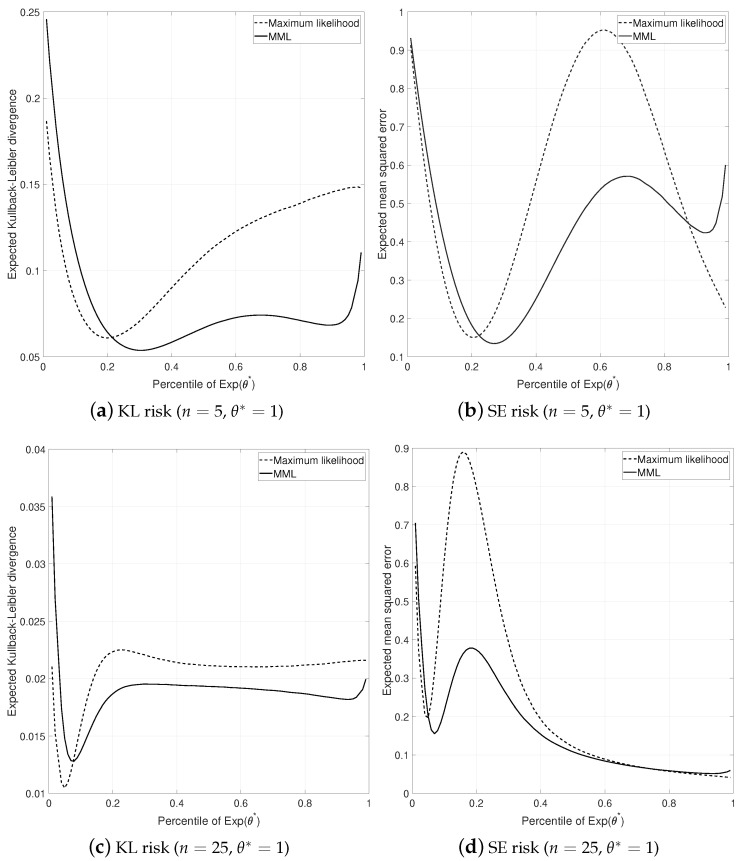
Expected Kullback-Leibler (KL) divergence and squared error (SE) risk of the maximum likelihood and MML87 estimates for n=5 (**top**) and n=25 (**bottom**) data points generated from model $\theta^{*}=1$. The *x*-axis on all plots denotes the censoring point *c* and is set to a percentile of Exp($\theta^{*}$).

**Table 1 entropy-23-01439-t001:** MML finite mixture models for Crime and Melanoma data. The attribute modelling censored survival time is seven in the Crime data set and five in the Melanoma data set.

Data	Class	Attributes						
		1	2	3	4	5	6	7
Crime	1	(50%, 50%)	(73%, 27%)	( 3%, 97%)	(42%, 58%)	(*r*: 2.0, *p*: 0.4)	(μ: 20.7, σ: 2.1)	(θ**: 119.0**)
	2	(55%, 45%)	(15%, 85%)	(23%, 77%)	(28%, 72%)	(*r*: 13.4, *p*: 0.8)	(μ: 24.9, σ: 3.3)	(θ**: 249.3**)
	3	(40%, 60%)	(16%, 84%)	(18%, 82%)	(51%, 49%)	(*r*: 2.3, *p*: 0.5)	(μ: 36.3, σ: 4.9)	(θ**: 345.9**)
Melanoma	1	(43%, 57%)	(17%, 83%)	(μ: 57.3, σ: 17.1)	(μ: 5.4, σ: 3.4)	(α**: 12.0**, β**: 7.3**)	–	–
	2	(73%, 27%)	(80%, 20%)	(μ: 49.5, σ: 15.8)	(μ: 1.4, σ: 0.9)	(α**: 8.1**, β**: 37.0**)	–	–

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
