# Peer review of "Minimum Message Length Inference of the Exponential Distribution with Type I Censoring"

_entropy, 2021, doi:10.3390/e23111439_

Round 1

Reviewer 1 Report

In their manuscript, the authors demonstrate how minimum message length can be used to infer data with Type I and fixed censoring information. The results reported represent an advance to the state of the art but that could be better explored. In my opinion, the manuscript is suitable for publication, after
the authors have addressed the following comments and questions:

1) highlight the scientific contribution explaining its relevance. It is not enough to say that it has never been demonstrated as it may never have been demonstrated because it does not make sense or its application has no relevance.

2) the references, although important, are all very old which brings me to the previous point. I have no doubts about the contribution, but about its relevance. It would be important to cite more recent articles.

3) The last section of the discussion - "6. Discussion" turns out to be the worst section of the article where in fact there is no discussion there is a very simple conclusion when summarizing the article and points to the future. It would be important to actually have a discussion in this section about the results that were presented.

Author Response

We thank the reviewer for their comments and suggestions which have no doubt improved the manuscript.

  • 1) highlight the scientific contribution explaining its relevance. It is not enough to say that it has never been demonstrated as it may never have been demonstrated because it does not make sense or its application has no relevance.

We have re-written the Discussion section to highlight the contributions of the manuscript. We have also added a paragraph in the Introduction that lists the key contributions of the manuscript.

  • 2) the references, although important, are all very old which brings me to the previous point. I have no doubts about the contribution, but about its relevance. It would be important to cite more recent articles.

We have included more recent citations, for example:

Kohjima, M.; Matsubayashi, T. & Toda, H. Variational Bayes for Mixture Models with Censored Data. Machine Learning and Knowledge Discovery in Databases, Springer International Publishing, 2019, pp. 605-620.

Mera, B.; Mateus, P. & Carvalho, A. M. On the minmax regret for statistical manifolds: the role of curvature. 2020

Dauda, K. A.; Pradhan, B.; Shankar, B. U. & Mitra, S. Decision tree for modeling survival data with competing risks, Biocybernetics and Biomedical Engineering, Elsevier BV, 2019, 39, 697-708

Balakrishnan, N. & Davies, K. F. Pitman closeness results for Type-I censored data from exponential distribution. Statistics & Probability Letters, Elsevier BV, 2013, 83, pp. 2693-2698.

  • 3) The last section of the discussion - "6. Discussion" turns out to be the worst section of the article where in fact there is no discussion there is a very simple conclusion when summarizing the article and points to the future. It would be important to actually have a discussion in this section about the results that were presented.

We have re-written the discussion section as per the suggestion of the reviewer. The section now includes: (i) a summary of the main contributions of the paper; (ii) a discussion of the key findings, (iii) an example of how the proposed codelengths can be used in more complex models (finite mixture modelling) and (iv) a comparison of MDL and MML philosophies of inference.

Reviewer 2 Report

Dear Colleagues, 

the paper seems to be written too fast as far as it contains too many typos. For example in the 2-nd row in the introduction "t" have to be "T". In the same row "c" have to be "C". In formula (6) the first lower index "T" have to be replaced with "Y". There is a typo also in the last formula in row 41, in the formula for the expectation. Please correct it and check all your work carefully for typos.

The well-known facts are better to be deleted. For example, in the beginning of Section 2, the definition of Exponential distribution is very well-known. May be you have you have written it to clarify the parameter, however it is better just to write in (3) the expectations of these random variables. 

The description of the references have to be carefully checked and corrected. For example, [5] and [6] have no year.

Author Response

We thank the reviewer for their comments and suggestions which have no doubt improved the manuscript.

  • The paper seems to be written too fast as far as it contains too many typos.

We have carefully read through the manuscript and hopefully fixed all typos.

  • For example in the 2-nd row in the introduction "t" have to be "T". In the same row "c" have to be "C".

We have made the suggested change: “In Type I random censoring we observe for each item $i$ either the true survival time $T_i = t_i$ ($t_i > 0$) or the censoring time $C_i = c_i$ ($c_i > 0$), where capital letters denote random variables.”

  • In formula (6) the first lower index "T" have to be replaced with "Y".

This has now been fixed.

  • There is a typo also in the last formula in row 41, in the formula for the expectation.

This has now been fixed. The summation index in the expectation of k^(a) is now set to k.

  • The well-known facts are better to be deleted. For example, in the beginning of Section 2, the definition of Exponential distribution is very well-known. May be you have you have written it to clarify the parameter, however it is better just to write in (3) the expectations of these random variables.

We have removed the definition of the PDF and CDF of the exponential distribution as per the reviewer’s suggestion. We have also removed: (1) the equation for the marginal probability of a single datum from Section 2; (2) the first equation in Section 2.1 that specifies the joint probability of the data; (3) the likelihood function for type I censored data in Section 2.1;

  • The description of the references have to be carefully checked and corrected. For example, [5] and [6] have no year.

The missing dates have been added to references [5] and [6]. We have also checked that the remaining references are correct and contain all the required information.

Reviewer 3 Report

The research work is interesting.  Authors need to read the entire draft for correcting some possible typos or adding some reference to make equations obtained readers better understood.  I have listed some of the items below. 

Line 21, "citeWallace05" should be changed.

Line 66, "for small sample sizes and problems"?

Remarks:  I understand "small sample sizes"; but I do not understand what problems considered.

Except Eq. (44) and Eq. (26), authors used uniform prior for the discussion.  Please provide some possible remarks at the conclusion section about using non-uniform prior. 

Provide some reference to get Eq. (34) and Eq. (35). 

Author Response

We thank the reviewer for their comments and suggestions which have no doubt improved the manuscript.

  • The research work is interesting.  Authors need to read the entire draft for correcting some possible typos or adding some reference to make equations obtained readers better understood.  I have listed some of the items below. 

We have carefully read through the manuscript and hopefully fixed all typos.

  • Line 21, "citeWallace05" should be changed.

This has been fixed and is now a proper citation.

  • Line 66, "for small sample sizes and problems"? Remarks: I understand "small sample sizes"; but I do not understand what problems considered.

We have re-worded this paragraph to include examples: “Furthermore, in models where the number of parameters grows with $n$ or the sample size is relatively small, the difference between the MML87 codelength and BIC can be substantial. Examples include analysis of multiple short time series, where several measurements are collected over a period of time for a large number of study participants~\cite{SchmidtMakalic16a}, and discriminating between Poisson and geometric distributions based on observed data~\cite{WongEtAl18}. In the latter example, both the Poisson and geometric distribution have the same number of free parameters so that model selection with BIC is equivalent to choosing the model with the higher likelihood. In contrast, MML87 takes into account the complexity of each distribution~\cite{Balasubramanian05} and not just the number of parameters, resulting in improved model selection performance for small sample sizes~\cite{WongEtAl18}.”

  • Except Eq. (44) and Eq. (26), authors used uniform prior for the discussion. Please provide some possible remarks at the conclusion section about using non-uniform prior.

We have re-written the Discussion section which now has: (i) the main contributions of the paper; (ii) an extension of the proposed codelengths to finite mixture models; and (iii) comparison to the closely-related MDL principle. We  also include a brief discussion of prior distributions and their effect on estimates.

  • Provide some reference to get Eq. (34) and Eq. (35).

Equations (34) and (35) are derived by the authors. We are not aware of other work that has the same estimate and derives the corresponding MSE. We have omitted the mathematical details (ie, integration w.r.t. a gamma distribution) from the paper as these are tangential to the aim of the paper but are happy to include them if required.

Round 2

Reviewer 1 Report

The authors improved the manuscript according to all the comments I made in the previous review

This manuscript is a resubmission of an earlier submission. The following is a list of the peer review reports and author responses from that submission.